# CHD4 regulates platinum sensitivity through MDR1 expression in ovarian cancer: A potential role of CHD4 inhibition as a combination therapy with platinum agents

Yoshiko Oyama[1], Shogo Shigeta[1], Hideki Tokunaga[1]*, Keita Tsuji[1], Masumi Ishibashi[1], Yusuke Shibuya[1], Muneaki Shimada[1], Jun Yasuda[2], Nobuo Yaegashi[1]

1 Department of Obstetrics and Gynecology, Tohoku University Graduate School of Medicine, Sendai, Japan, 2 Division of Molecular and Cellular Oncology, Miyagi Cancer Center Research Institute, Natori, Miyagi, Japan

* hideki.tokunaga.a1@tohoku.ac.jp

**Data Availability Statement:** A part of the data underlying the results presented in the study are available from cBioPortal (https://www.cbioportal. org/ ). The other relevant data are all contained

## Abstract

Platinum sensitivity is an important prognostic factor in patients with ovarian cancer. Chromo-domain-helicase-DNA-binding protein 4 (CHD4) is a core member of the nucleosome remodeling and deacetylase complex, which functions as a chromatin remodeler. Emerging evidence indicates that CHD4 could be a potential therapeutic target for cancer therapy. The purpose of this study was to clarify the role of CHD4 in ovarian cancer and investigate its therapeutic potential focusing on platinum sensitivity. In an analysis of the Cancer Genome Atlas ovarian cancer dataset, CHD4 gene amplification was associated with worse overall survival. *CHD4* mRNA expression was significantly higher in platinum-resistant samples in a subsequent clinical sample analysis, suggesting that CHD4 overexpression conferred platinum resistance to ovarian cancer cells, resulting in poor patient survival. In concordance with these findings, CHD4 knockdown enhanced the induction of apoptosis mediated by cisplatin in ovarian cancer cells TOV21G and increased cisplatin sensitivity in multiple ovarian cancer cells derived from different subtypes. However, CHD4 knockdown did not affect the expression of RAD51 or p21, the known targets of CHD4 in other cancer types that can modulate platinum sensitivity. Knockdown and overexpression assays revealed that CHD4 positively regulated the expression of multi-drug transporter MDR1 and its coding protein p-glycoprotein. In addition, a first-in-class CHD4/SMARCA5 inhibitor ED2-AD101 showed synergistic interactions with cisplatin. Our findings suggest that CHD4 mediates platinum sensitivity by modulating MDR1 expression in ovarian cancer. Further, CHD4 suppression has a potential to be a novel therapeutic strategy in combination with platinum agents.

## Introduction

More than half of the ovarian cancers are diagnosed in advanced stages, and present with carcinomatous peritonitis [1]. The therapeutic strategy for advanced ovarian cancer is based on a combination of chemotherapy and debulking surgery [2]. Although several molecular-targeted

within the manuscript and its Supporting Information files. If needed, raw data related to this paper can be provided upon reasonable request to the corresponding author.

**Funding:** This study was financially supported by the JSPS KAKENHI Grants (19H03795 to NY, 19K09747 to KT, 19K23904, and 20K18211 to SS). https://www.jsps.go.jp/english/e-grants/index.html The funders had no role in study design, data collection and analysis, decision to publish, or preparation of the manuscript.

**Competing interests:** The authors have declared that no competing interests exist.

drugs such as VEGF antibodies and PARP inhibitors have become available, for more than a decade frontline chemotherapy has been based on platinum agents, regardless of the subtype [3–5]. Therefore, the sensitivity to platinum agents is the most important prognostic factor for patient survival [6].

Platinum agents are the major DNA damaging agents that exert anticancer effects mainly by forming intrastrand and interstrand DNA crosslinks [7,8]. Although about 70% of the patients with ovarian cancer initially respond to platinum-based chemotherapy, some develop resistance during the primary treatment [9,10]. Recurrence after remission is almost inevitable in most of the other cases, and patient prognosis is poor once they have developed platinum resistance [9,11]. Thus, a strategy to overcome platinum resistance is an important concern in ovarian cancer treatment.

Chromodomain-helicase-DNA-binding protein 4 (CHD4) is a component of the nucleo-some remodeling and deacetylase (NuRD) complex. As the name suggests, NuRD complex acts as a chromatin remodeler and CHD4 is considered to play a pivotal role by conferring ATPase activity on the complex [12–14]. The NuRD complex is believed to act as a transcription repressor, but recent research indicated that it can also promote gene transcription globally [14,15]. NuRD or CHD4 is reported to be involved in several important biological functions in cancer cells, such as cell differentiation, cell cycle regulation, and DNA damage repair [16–22]. It has also been reported that CHD4 expression was correlated with patient prognosis in a variety of malignancies [23–26]. These findings suggest that CHD4 could be a potential therapeutic target in cancer therapy. However, the importance of CHD4 in ovarian cancer has not been fully studied.

The purpose of this study was to elucidate the potential of CHD4 as a new therapeutic target for the treatment of ovarian cancer, with a particular focus on its interaction with platinum agents.

## Materials and methods

### Cell lines and cell culture conditions

TOV21G was obtained from the American Type Culture Collection. KURAMOCHI was obtained from the Japanese Cancer Research Resources Bank. JHOS2, and JHOC5 were obtained from RIKEN. A2780 and A2780cis were obtained from Sigma-Aldrich. OSE2 and OSE4 were kindly provided by Dr. Makoto Nitta (University of Kumamoto, Kumamoto, Japan) [27]. TOV21G and KURAMOCHI were maintained in RPMI1640 with 10% fetal bovine serum (FBS). JHOS2 and JHOC5 were cultured in Dulbecco's modified Eagle's medium Nutrient Mixture F-12 (DMEM/F-12) supplemented with 10% FBS and MEM non-essential amino acid. A2780, A2780cis, OSE2, and OSE4 were cultured in DMEM supplemented with 10% FBS. All cells were cultured in 5% CO2 at 37˚C. All of the cells were validated as mycoplasma negative at the time of arrival and cell aliquots were cryopreserved after a couple of passages for subsequent experiments.

### TCGA data analysis

Data on genomic alterations and patient survival among 489 ovarian cancer patients registered to The Cancer Genome Atlas Network (accession number: phs000178) were downloaded via cBioPortal (https://www.cbioportal.org/) [28,29]. Overall survival was analyzed using the Kaplan-Meier method and survival difference was compared using the log-rank test.

### Clinical patient data analysis

Histologically confirmed primary ovarian high-grade serous or clear cell carcinoma specimens from patients who underwent surgery at Tohoku University Hospital between January 2011

and December 2017 were subjected to analysis with waived informed consent under the approval of the ethics committee at Tohoku University Graduate School of Medicine (approval number: 2020-1-238.). The investigator accessed the medical record and the stored samples during August 2020 to September 2020. The data were not fully anonymized before the investigators accessed them. To assess CHD4 RNA expression, only those patients whose specimens were stored at -80°C after embedding in OCT compound were eligible (Sakura Finetechnical Co. Ltd., Tokyo). Samples were homogenized, and RNA was extracted and quantified as described later. Clinical information, including overall survival and platinum sensitivity, were also collected. In this study, a specimen was considered to be a platinum-sensitive one only if clinically detectable tumor remained after the surgery and the patients responded to the subsequent three cycles of platinum-based chemotherapy immediately after the surgery. In contrast, a specimen was considered to be platinum resistant if the detectable tumor showed progressive or non-regressive behavior toward subsequent chemotherapy after primary suboptimal surgery. The platinum sensitivity was considered to be unknown in cases where the tumor was completely or almost completely resected by surgery. The correlation between CHD4 mRNA expression with overall survival and platinum sensitivity was assessed using the log-rank test and Mann-Whitney test, respectively.

## Immunohistochemistry

Formalin-fixed, paraffin-embedded tissue sections were treated with 3% hydrogen peroxide diluted in methanol after deparaffinization and heat-induced epitope retrieval. Tissue sections were then incubated with Anti-CHD4 antibody (ab70469, Abcam), which was diluted to the concentration of 1:200 at 4°C overnight, and subsequently treated with Histofine® Simple Stain MAX-PO(M) (Nichirei) at room temperature for 30 minutes. Antigen-antibody complex was visualized with 3,3‾diaminobenzidine solution. After counterstaining with hematoxylin, immunohistochemical reaction was evaluated through a light microscope.

## CHD4 short interfering RNA (siRNA) transfection

Cells were transfected with either 5nM of pooled siRNA targeting CHD4 or negative control using Lipofectamine RNAiMAX (Thermo Fisher Scientific) according to the manufacturer's instructions. siRNAs for CHD4-1 (#s2984), CHD4-2 (#s2985), and negative control siRNA (#4390843) were purchased from Thermo Fisher Scientific.

## RNA extraction, cDNA synthesis, and quantitative RT-PCR

Total RNA was extracted using the RNeasy Mini Kit (QIAGEN) or ISOGENII (NIPPON GENE). Reverse transcription was performed using ReverTra Ace qPCR RT Master Mix with gDNA Remover (FSQ-301, TOYOBO) to obtain complementary DNA. Quantitative PCR was performed using the StepOne Plus Real-Time PCR System with TaqMan® Fast Advanced Master Mix (Applied Biosystems) and a TaqMan probe specific for each gene. Relative gene expression was measured using the ΔΔCT method. The expression of *CHD4*, *ABCC1*, *ABCC2*, and *MDR1* was normalized to *GAPDH* mRNA expression. The following TaqMan probes were used in this study: *CHD4* (Hs00172349_m1), *MDR1* (ABCB1) (Hs00184500_m1), *ABCC1* (Hs01561512_m1), *ABCC2* (Hs00166123_m1), and *GAPDH* (NM_002046.3) (all from Thermo Fisher Scientific).

## Protein extraction and Western blot analysis

Proteins were extracted from cells using M-PER mammalian protein extraction reagent (#78501, Thermo Fisher Scientific), and protein concentration in each sample was quantified

using a BCA Protein Assay kit (# 23227 Thermo Fisher Scientific). Samples were subjected to sodium dodecyl sulfate poly-acrylamide gel electrophoresis (SDS-PAGE) using 5–20% Super-Sep (#194–15021, FUJIFILM) and transferred to a PVDF membrane at 180 mA for 90 min. After blocking with 5% skimmed milk in TBST (10 mM Tris–HCl, pH 7.6, 150 mM NaCl, 0.2% Tween-20) for 1 h, membranes were incubated with primary antibody in Can Get Signal® Immunoreaction Enhancer Solution1. Primary antibodies were detected using anti-mouse (#NA931, Cytiva) or anti-rabbit (#7074S Cell Signaling Technologies) HRP-conjugated IgG diluted 1:10000 in. Can Get Signal® Immunoreaction Enhancer Solution2 (NKB-101, TOYOBO). Blots were developed using ECL Prime Western Blotting Detection Reagent (RPN2232, Cytiva) and detected by Chemidoc (Bio-Rad). The following antibodies were used for Western blotting: Anti-CHD4 antibodies from Abcam (ab70469), anti-PARP antibodies (#9542), anti-Phospho-Histone H2A.X antibodies, (#9718), and anti-RAD51 antibodies (#8875) from Cell Signaling Technologies; anti-P-glycoprotein antibodies (#108370) and anti-p21 antibodies (#100444) from GeneTex. Anti-beta-actin antibodies (NB600-501) from Novus. All antibodies were diluted 1:1000 except for anti-beta actin antibodies, which were used at 1:5000 dilution. Densitometry was performed using Fiji software. [30]

## Cell viability assay

About 2000–5000 cells were plated in 96-well plates. After siRNA transfection or inhibitor treatment, cell viability was assessed using the Cell Counting Kit-8 (DOJINDO) according to the manufacturer's protocol. The absorbance in each well at 450 nm was measured using an iMark™ Microplate Absorbance Reader (Bio-Rad).

**Clonogenic assay.** After washing with phosphate buffered saline (PBS), cells plated on a 6-well plate were fixed with 100% methanol at -20˚C for 10 minutes. The cells were then stained with 0.5% crystal violet for 10 minutes at room temperature.

## Flow cytometry

After the treatment, cells were washed with PBS and stained in Annexin V Binding Buffer with SYTOX™ AADvanced™ Dead Cell Stain Kit (S10349, Thermo Fisher Scientific) and annexin V, Pacific Blue conjugate (A35122, Thermo Fisher Scientific) following the manufacturer's instructions. The fraction of Annexin V-positive and Pacific blue positive cells were measured using flow cytometry with FACS Canto™ II (BD) and analyzed using FACS Diva software (BD).

## Cell cycle analysis

After the treatment, cells were washed with PBS, fixed with 70% ethanol, and stained with Propidium Iodide (P3566, Thermo Fisher Scientific). Cell cycle analysis was performed by flow cytometry with FACS Canto™ II and analyzed by FACS Diva software.

## Drug interaction assessment using Bliss independence model

The SMARCA5/CHD4 inhibitor ED2-AD101 was kindly provided by Dr. Saunthararajah [31]. Cells were plated in 96-well plates 24 h prior to treatment. Cells were treated with the designated concentrations of ED2-AD101 and cisplatin. Cell viability was assessed 72 h after drug administration and cell viability was determined as mentioned above. The percentage of cell growth inhibition compared to mock treatment was determined. A Bliss independence model was employed to evaluate the drug interaction effects [32]. Briefly, the Bliss independence model was used to estimate growth inhibition if the two agents were independent. When the cell growth inhibition (%) by each drug A or B is EA and EB, respectively, the Bliss index

(Ebliss), which is the calculated value when the combined action of drugs A and B is an additive effect, is calculated using the following formula: Ebliss = E + EB (EA×EB). In this study, the difference between the observed growth inhibition and expected growth inhibition, which was calculated using the Bliss independence model was defined as the Bliss independence value (BI value). BI values of more than 0 and less than 0 suggest antagonistic and synergistic interactions, respectively.

## Vector transfection

TOV21G cells were plated as $3×10^5$ cells per well on a 6-well plate. After 24 h, the cells were transfected with 1 μg/well of human CHD4 overexpression vector (FHC11086 from Kazusa DNA Research Institute) or the corresponding pFN21A-and-pFN21K-HaloTag-CMV-Flexi empty vector (G282A from Promega) using Lipofectamine 2000 transfection reagent (Life Technologies) according to the manufacturer's instructions.

## Statistical analysis

Statistical analyses were performed using GraphPad Prism 8.0 (GraphPad Software Inc.). If not mentioned otherwise, the values in the figures represent the means of three independent assays. Error bars in each figure represent the standard deviation (SD). Statistical analysis was performed using Student's $t$-test, except for the log-rank test and Mann-Whitney test in the clinical data set analyses. P-values less than 0.05 were considered statistically significant.

## Results

### CHD4 amplification or overexpression is associated with poor survival in patients with ovarian cancer

To investigate the role of *CHD4* in the outcome among patients with ovarian cancer, we first referred to the TCGA dataset and analyzed the relationship between *CHD4* gene alterations and overall survival of the patients. Mutations in *CHD4* were found in 2.5% of the patients, and we noticed that a small number of patients harbored *CHD4* amplification. Among the 485 cases, in which the information on putative copy number was available, 27 cases had *CHD4*

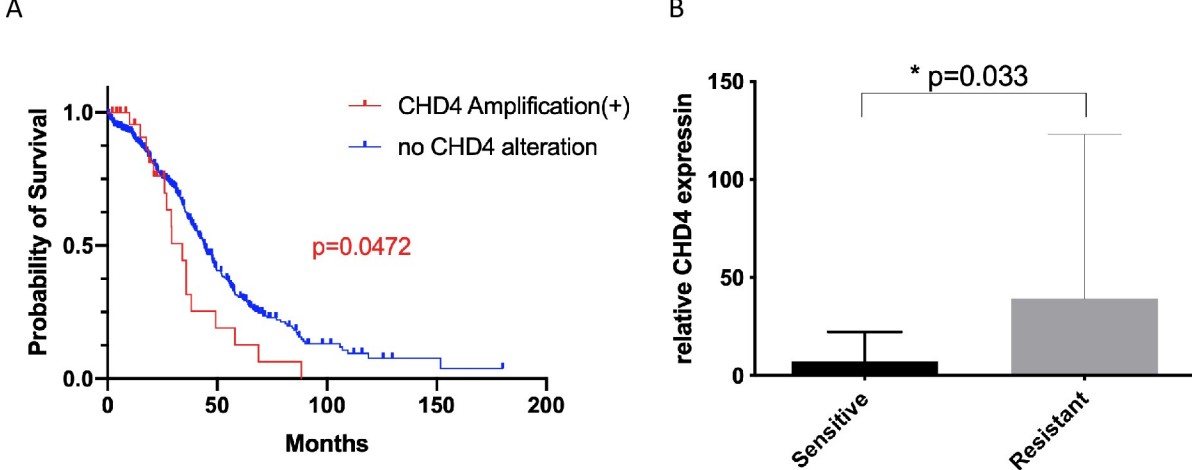

**Fig 1. CHD4 amplification or higher expression is associated with worse survival or platinum sensitivity in patients with ovarian cancer.** (A) Comparison of overall survival among the patients registered in the TCGA project with or without *CHD4* amplification. Red and Blue curves represent overall survival in the patients harboring *CHD4* amplification (n = 27) and the patients without *CHD4* copy number alteration (n = 458), respectively. (B) *CHD4* mRNA expression in platinum sensitive (n = 20) and platinum resistant (n = 6) patient samples.

amplification while no deletion was detected. Importantly, patients harboring *CHD4* amplification showed significantly worse overall survival than those without *CHD4* copy number alteration. (p = 0.0475) (Fig 1A).

To verify the above findings in our clinical database, we accessed the clinical dataset and tissue bank at the Tohoku University Hospital. We focused on high-grade serous cancer and clear cell carcinoma as these are the most and second most frequent histologic subtypes in our nation. Patient characteristics are shown in Table 1. Twenty-nine patients with high-grade serous carcinomas and 32 with clear-cell carcinomas were enrolled. Based on the definition described above, 20 cases were considered as platinum sensitive, 6 as resistant, and 35 as unknown platinum sensitivity. As platinum-resistance is one of the most critical prognostic factors in patients with ovarian cancer, the information on platinum sensitivity as well as on overall survival in each patient was obtained. Patient characteristics are summarized in Table 1. RNA was extracted from the stored samples and *CHD4* mRNA expression was quantified using qPCR. With regard to patient survival, there was no statistically significant difference between the two groups divided at median *CHD4* mRNA expression. (p = 0.084, S1A Fig). Interestingly, however, *CHD4* expression was significantly higher in platinum-resistant cases (p = 0.033), suggesting that *CHD4* overexpression had the potential to lead platinum resistance in patients with ovarian cancer (Fig 1B). Immunohistochemistry was additionally performed to further investigate the interaction between CHD4 expression and platinum sensitivity. Majority of the specimens were positive for CHD4 immunostaining. Interestingly, the six platinum resistant samples were all positive for CHD4 expression, and all of the samples negative for CHD4 immunostaining belonged to the platinum sensitive group although no statistically significant difference was observed. (S1B and S1C Fig). The results from the clinical data analysis led us to hypothesize that *CHD4* expression possibly influences patient prognosis by regulating platinum sensitivity in ovarian cancer.

## CHD4 knockdown restores the sensitivity to cisplatin in ovarian cancer cells

To experimentally prove the hypothesis that *CHD4* regulates platinum sensitivity in ovarian cancer, ovarian cancer cell lines were subjected to *in vitro* experiments. First, the expression of CHD4 in several ovarian cancer cell lines was compared by quantitative PCR and Western blotting to examine whether there was a significant difference among ovarian cancer cells including A2780 and A2780cis, a pair of parental cells and experimentally acquired platinum-resistant derivatives. Although quantitative PCR indicated that CHD4 expression tends to be higher in KURAMOCHI, A2780, and A2780cis, obvious difference was not identified across the cell lines by Western blotting (S2 Fig). Thus, we decided to select two clear-cell subtype cells (TOV21G, JHOC5) and two high-grade serous subtype cells (JHOS2, KURAMOCHI), the biomolecular characteristics of whose original subtypes are well conserved [33–35]. The

**Table 1. Characteristics of ovarian cancer patients.**

| Histological type | Serous | 29 |
|---|---|---|
| | Clear | 32 |
| Clinical stage (FIGO2014) | I | 23 |
| | II | 9 |
| | III | 21 |
| | IV | 8 |
| Platinum Sensitivity | Sensitive | 20 |
| | Resistant | 6 |
| | N.A | 35 |

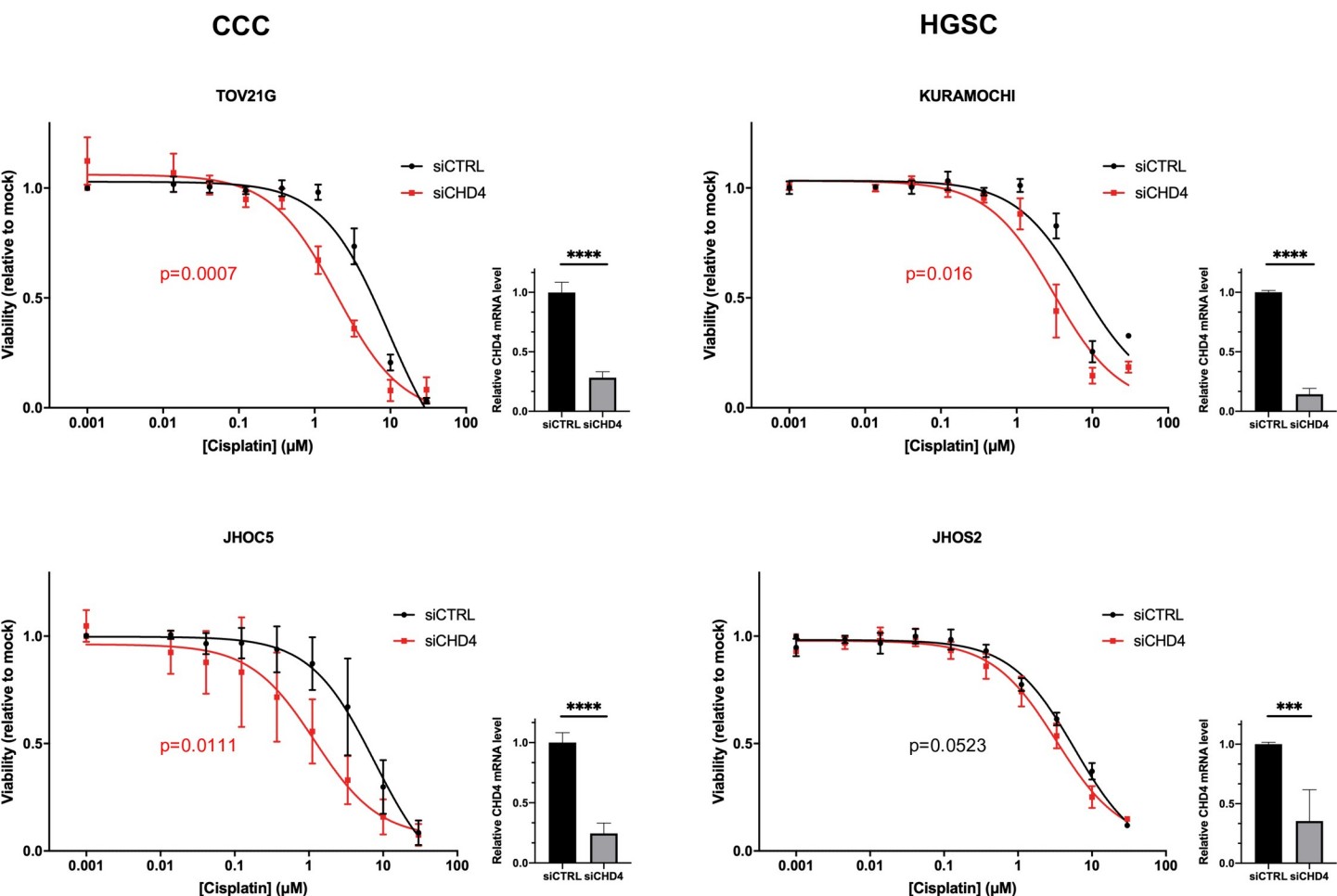

**Fig 2. Inhibition of CHD4 expression increases platinum sensitivity in ovarian cancer cell lines.** Dose-response curves for 72 hours of cisplatin treatment in TOV21G, JHOC5, KURAMOCHI, and JHOS2 after 72 hours of *CHD4* siRNA or negative control siRNA transfection. The curves represent the average values of at least three independent experiments. IC50 between samples with *CHD4* siRNA transfection and control siRNA transfection was compared using Student's *t*-test. The efficiency of *CHD4* knockdown was validated using quantitative PCR in all the cells. ***$p \le 0.001$; ****$p \le 0.0001$. siCTRL; negative control siRNA.

cells were treated with the designated concentrations of cisplatin for 72 h after transfection either with siRNA targeting *CHD4* or negative control. The efficiency of *CHD4* knockdown was verified using quantitative PCR in all cells. Cell viability was assessed 72 h after cisplatin administration to obtain dose-response curves and determine IC50 (Fig 2 and S1 Table). Of the four cell lines, *CHD4* knockdown significantly decreased the IC50 of cisplatin in TOV21G, JHOC5, and KURAMOCHI. While the IC50 was not significantly different in the three independent assays, the same trend was observed in JHOS2 (Fig 2 and S1 Table). To validate the influence of CHD4 knockdown on cisplatin sensitivity in detail, TOV21G, JHOC5 and KURA-MOCHI were treated with 0.5, 1, 2, 5, and 10 µM of cisplatin. CHD4 knockdown significantly increased the sensitivity to cisplatin at the multiple concentration points as demonstrated in S3 Fig. In concordance with our hypothesis, the results revealed that *CHD4* expression regulated platinum sensitivity in ovarian cancer cells. Clonogenic assay further validated that CHD4 knockdown enhances the suppressive influence of cisplatin on colony formation (S4 Fig). OSE2 and OSE4, two immortalized cell lines derived from normal ovarian surface epithelium, were additionally tested to compare the difference between cancer and non-cancerous cells. The sensitivity to cisplatin did not significantly differ between CHD4 knockdown and control

in these non-cancerous cell lines, which indicates the cancer specific involvement of CHD4 in the regulation of platinum sensitivity (S5 Fig). To further understand the direct influence of *CHD4* knockdown on cell viability independent of cisplatin treatment, the cell viability was compared after 72 to 120 h after *CHD4* knockdown in the four ovarian cancer cells. The response varied among cell lines, indicating that the combination of treatment with platinum agents and *CHD4* suppression was a more promising therapeutic strategy for ovarian cancer than *CHD4* inhibition alone (S6 Fig).

## Novel SMARCA5/CHD4 inhibitor ED2-AD101 synergizes with cisplatin

Although not specific to CHD4, Dr. Saunthararajah and his colleagues established a SMARCA5/ CHD4 dual inhibitor ED2-AD101 that potently suppressed AML cell growth in cell-based assays [31], which motivated us to investigate whether ED2-AD101 increased the sensitivity of ovarian cancer cells to platinum agents. IC20 of ED2-AD101 for each cell was preliminarily determined during 72 h of dose-response assays (5 μM for TOV21G and KURAMOCHI, 7.5 μM for JHOC5; S7 Fig). As it was difficult to determine IC20 for JHOS2 based on the dose response curve, which was the least sensitive cell line to ED2-AD101 treatment, 10 uM of ED2-AD101 was administered to JHOS2 instead. These cells were treated with the designated concentrations of cisplatin in the presence of IC20 of ED2-AD101 or vehicle for 72 h. Among the four cells, IC50 was significantly lower in TOV21G, JHOC5, and KURAMOCHI, consistent with the results shown in Fig 2. The increase of platinum sensitivity was not observed in JHOS2 (Fig 3A). To further assess the synergistic interaction of ED2-AD101 and cisplatin in detail, a Bliss independence (BI) model was employed. The left panels in Fig 3B show the observed percentage viability at the indicated concentrations of ED2-AD101 and cisplatin. The right panel shows the color-coded BI values, where red, white, and blue indicate synergistic, additive, and antagonistic interactions, respectively. The results clarified that ED2-AD101 and cisplatin demonstrated synergistic reactions at most of the points in TOV21G, JHOC5 and KURAMOCHI. Interestingly, synergistic interaction was also suggested at almost all concentration points in JHOS2 as well. The results with ED2-AD101 successfully validated the potential of CHD4 as a novel therapeutic target in ovarian cancer cells in combination with cisplatin.

## Inhibition of CHD4 enhances cisplatin-induced apoptosis

One of the major mechanisms of platinum agents in cancer therapy is to induce apoptosis by facilitating DNA damage, including DNA double-strand breaks [36,37]. It is also noteworthy that previous studies have shown that CHD4 was involved in DNA damage repair [12,17,21]. This suggests that CHD4 inhibition enhances DNA damage and subsequent apoptosis induced by platinum agents. To investigate whether CHD4 was involved in cisplatin-related apoptosis in ovarian cancer, the change in the phosphorylation at the histone H2AX at Serine 139 (γH2AX) and PARP cleavage were examined using Western blot analysis. TOV21G cells were transfected with either siRNA targeting CHD4 or negative control siRNA for 72 h. Subsequently, the cells were treated with 5μM cisplatin or vehicle for 24 h and protein was harvested. Whereas, *CHD4* knockdown alone did not affect the expression of γH2AX, and *CHD4* siRNA transfection significantly enhanced γH2AX overexpression induced by cisplatin treatment. With regard to PARP expression, *CHD4* knockdown unexpectedly increased both cleaved and non-cleaved forms in the presence of cisplatin, which makes it difficult to interpret these results (Fig 4A). Thus, flow cytometry was performed to assess the apoptosis. As shown in Fig 4B, *CHD4* knockdown significantly increased the fraction of Annexin-V positive cells compared to the control in the presence of cisplatin (34.1% ± 2.3 vs. 21.1% ± 5.3). In contrast, no significant difference was observed without cisplatin treatment. Taken together, these results

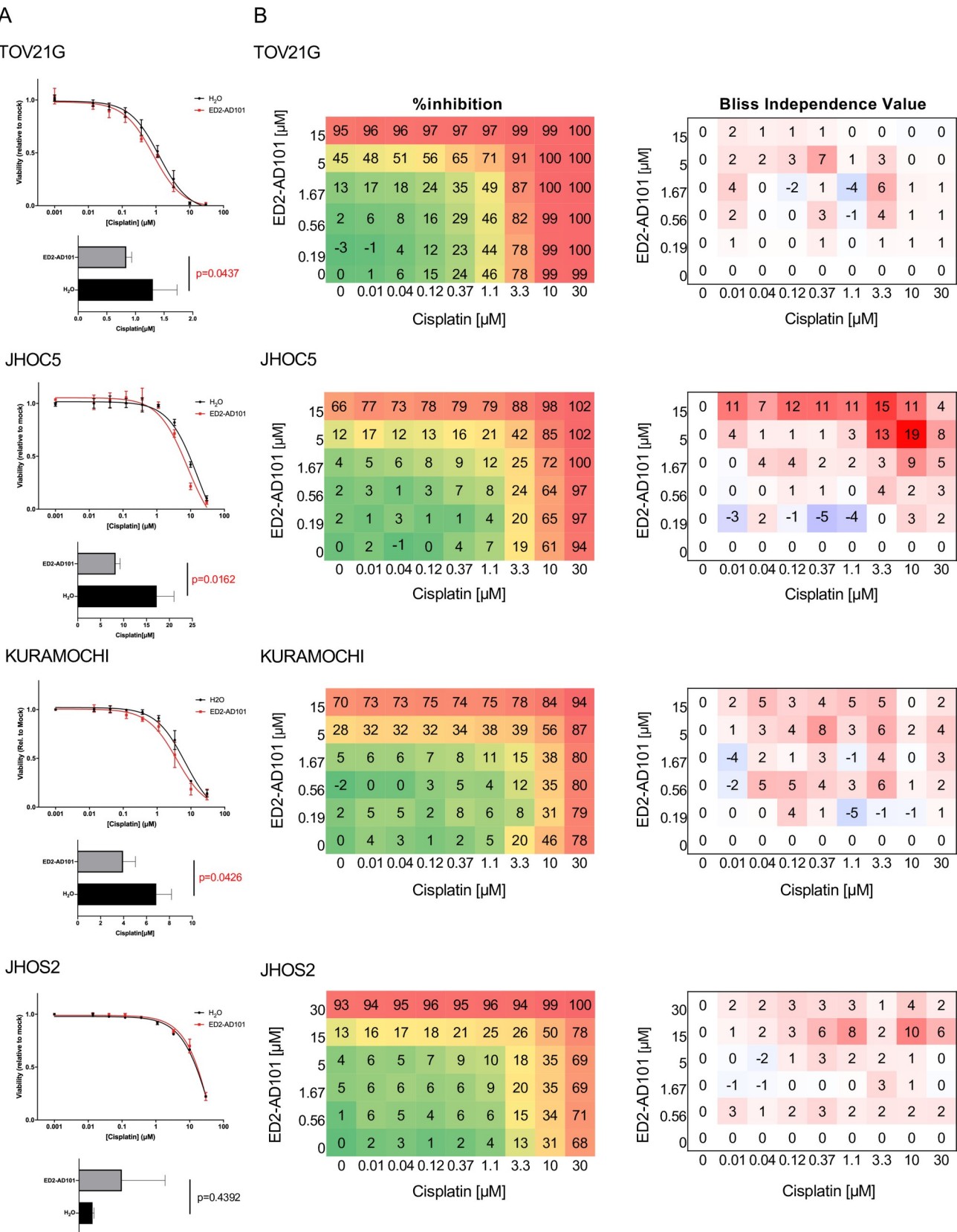

**Fig 3. CHD4 inhibitor ED2-AD101 shows synergistic interaction with cisplatin.** (A) TOV21G, JHOC5, KURAMOCHI, and JHOS2 cells were treated with 0–30 µM of cisplatin and IC20 of ED2-AD101 or vehicle for 72 hours. IC50 was compared using Student's *t*-test. (B) TOV21G, JHOC5, KURAMOCHI, and JHOS2 cells were treated with either the designated concentrations of cisplatin and ED2-AD101 or vehicle for 72 hours. Left panels demonstrate the observed percent growth inhibition relative to mock treatment. Right panel is the calculated bliss independence (BI) value. BI values less than and more than 0 indicate antagonistic and synergistic interactions, respectively. Each value is the mean of three independent assays.

suggest that CHD4 inhibition enhances cisplatin-mediated apoptotic cell death by promoting cytotoxic DNA damage in ovarian cancer cells.

Because cisplatin-induced apoptosis is accompanied by transient S phase and G2/M arrest [36], we further analyzed whether CHD4 inhibition interacts with cisplatin from the perspective of cell cycle control using cell cycle analysis. Approximately, 24 hours of cisplatin treatment strongly facilitated S phase arrest. CHD4 knockdown also led to obvious S phase arrest and mild G2/M arrest as well in TOV21G. Neither S phase nor G2/M phase arrest was synergistically enhanced by the combination of CHD4 knockdown and cisplatin treatment, suggesting CHD4 inhibition and cisplatin work on cell cycle regulation independently (S8 Fig).

## CHD4 modulates platinum sensitivity by positively regulating MDR1 expression

Variable factors modulate platinum sensitivity [38]. While it was not tested in ovarian cancer, in glioblastoma and breast cancers, it was demonstrated that knockdown of CHD4 decreases the expression of RAD51 and p21 that are tightly involved in cisplatin sensitivity [39–42]. Thus, we next examined the expression of RAD51 and p21 under *CHD4* knockdown in two ovarian cancer cells, TOV21G, and JHOC5. Different from our expectation, the expression of RAD51 or p21 did not decrease under CHD4 knockdown (Figs 4C and S9A). As RAD51 expression seemed to slightly increase in TOV21G instead, we quantified and statistically compared the relative RAD51 expression between control and CHD4 knockdown but did not find a significant difference (S9B Fig).

Another important factor related to cisplatin sensitivity is the function of multi-drug efflux transporters [38,43]. The influence of *CHD4* knockdown on the expression of several known multi-drug efflux transporter ATP-binding cassette (*ABC) C1*, *ABCC2*, and *ABCB1* or multi-drug resistance-1 (MDR1) were examined using quantitative PCR. Interestingly, the expression of *MDR1* was significantly decreased when *CHD4* expression was suppressed in TOV21G cells (Fig 4D). Western blot analysis confirmed that CHD4 knockdown suppressed P-glycoprotein expression in TOV21G as well as JHOG5 and JHOS2, suggesting that CHD4 was involved in platinum sensitivity by regulating drug efflux via P-glycoprotein in ovarian cancer. (Figs 4E and S9C).

To further confirm the involvement of CHD4 in *MDR1* transcription, the *CHD4* overexpression vector was introduced to TOV21G. Compared with the empty vector, the overexpression vector introduction increased the expression of *MDR1* along with an increase in *CHD4* expression, as expected (Fig 4F).

## Discussion

In this study, we clarified that CHD4 was involved in platinum sensitivity in ovarian cancer by positively regulating MDR1 expression. We have also successfully demonstrated the potential of the combination therapy of platinum agents and CHD4 inhibitor by introducing the first-in-class SMARCA5/CHD4 inhibitor ED2-AD101 in ovarian cancer cells.

There are several known biological mechanisms regulating platinum sensitivity [38]. Of these, the involvement of DNA repair systems such as nucleotide excision repair, mismatch

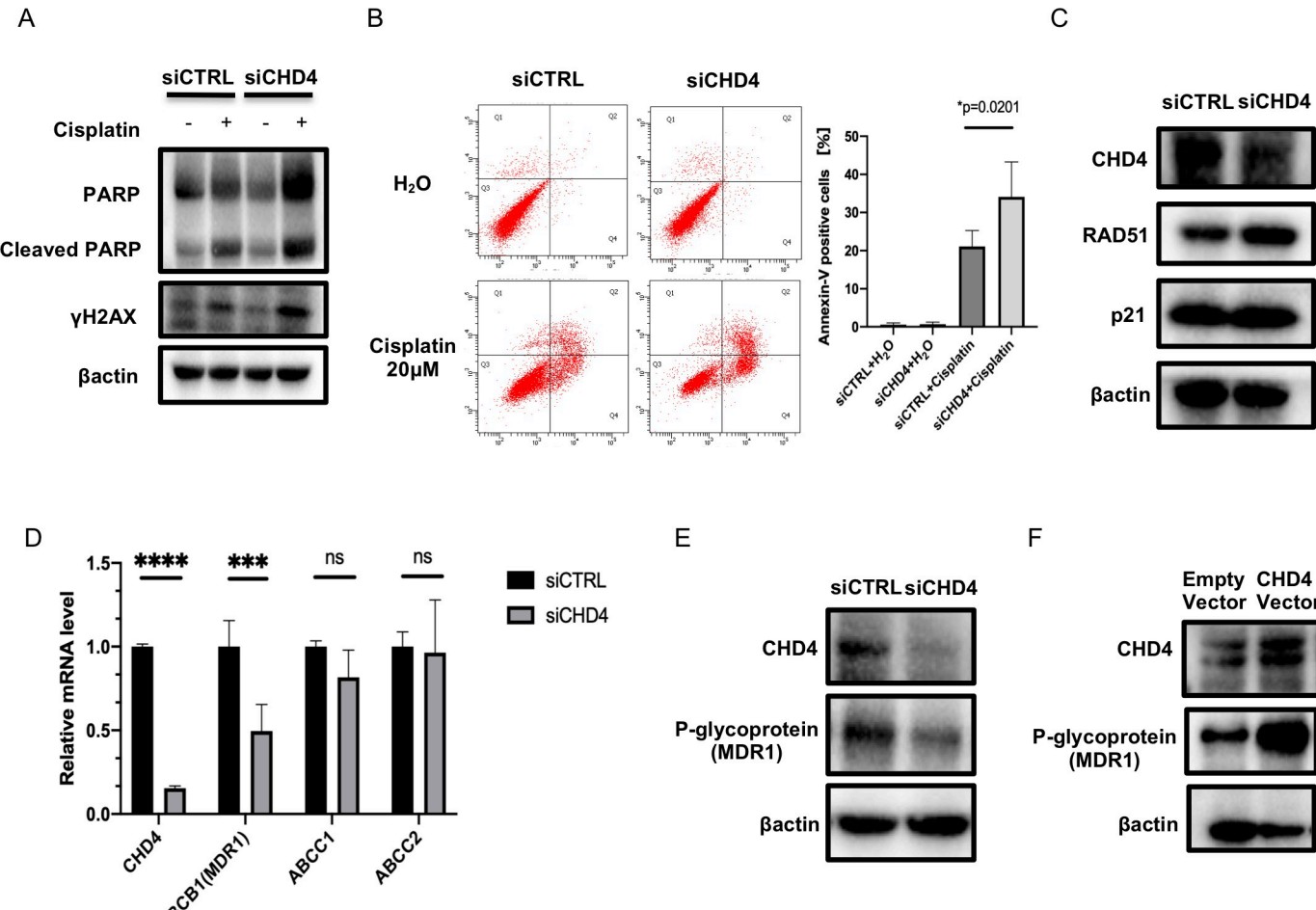

**Fig 4. CHD4 coordinates platinum-mediated apoptotic change by regulating MDR1 expression in ovarian cancer cells.** (A) After 72 hours of siRNA transfection, TOV21G cells were treated with either 5 μM of cisplatin or vehicle for 24 hours. Protein is harvested and the expression of cleaved PARP and γH2AX were examined using Western blot analysis. The experiment was repeated two times independently to confirm the reproducibility. (B) Apoptosis assay was analyzed using flow cytometry. After 72 hours of siRNA transfection, TOV21G cells were treated with either 20 μM of cisplatin or vehicle for 24 hours and the cells were subjected to flow cytometry. Left panel show the representative data. X-axis and Y-axis represent PerCP-Cy5.5 and annexin-V, respectively. Right panel summarizes the independent assays. Values represent the mean percentage of annexin-V positive cells. (C) The expression of known targets of *CHD4* in TOV21G. Protein was harvested after 72 hours of siRNA transfection. The expression of RAD51, p21 was examined using Western blot analysis. The blots are the representative data of three independent assays. (D) The influence of *CHD4* inhibition on multidrug efflux transporters *ABCC1*, *ABCC2*, and *MDR1* were investigated using quantitative PCR. RNA as harvested 48 hours after siRNA transfection. (E) Protein was harvested 72 hours after siRNA transfection in TOV21G and the expression of MDR1 was examined using Western blot analysis. The blots are the representative of two independent assays. (F) TOV21G cells were transfected with *CHD4* overexpression vector for 96 hours and P-glycoprotein as well as CHD4 expression was examined using Western blot analysis. The reproducibility of the result was validated by two independent assays. Abbreviations: siCTRL; negative control siRNA ns; not significant; $^*p \le 0.05$; $^{**}p \le 0.001$; $^{***}p \le 0.0001$.

repair, homologous recombination (HR), and non-homologous end binding has been well investigated [38,44]. For example, cisplatin sensitivity can be increased by inhibiting the function of the molecules implicated in the HR pathway, such as BRCA1, BRCA2, and RAD51 [45–47]. Dysfunction of p21, which leads to G1 cell cycle arrest in response to DNA damage in a p53-dependent manner, is also involved in HR and is known to increase platinum sensitivity [37,48,49]. Another important mechanism related to platinum sensitivity is the regulation of influx or efflux via multi-drug transporters [43]. Among these, P-glycoprotein encoded by MDR1 is one of the most studied cell multi-drug transporters that are highly expressed on the membrane of multidrug-resistant cancer cells [50,51]. As the name implies, MDR1/P-

glycoprotein is reported to be involved in the resistance of various drugs including anti-cancer agents such as paclitaxel, doxorubicin, PARP inhibitors, and importantly, cisplatin [52].

Contrary to our expectation, the expression of RAD51 or p21, which are the known targets of CHD4 as it was shown in other cancer types [40,53,54], was not clearly affected by CHD4 knockdown in the ovarian cancer cells we tested. In contrast, our experimental results and supporting literature indicated that reduced platinum efflux via P-glycoprotein was one of the mechanisms of action of CHD4 suppression that enhanced the effect of cisplatin. It should be noted that CHD4 possibly regulates platinum sensitivity through more variable and complex mechanisms. Further investigation is necessary to determine whether the regulation of MDR1 expression is a pivotal mechanism. At the same time, the results of this study imply the cancer type-specific function of CHD4. The common and cancer type-specific gene transcription regulation should also be further assessed to unveil the detailed function of CHD4 as well. Another question to be addressed is the mechanism of regulation of MDR1 expression by CHD4. MDR1 expression is complicatedly regulated by several important cancer-associated pathways such as RAS/Raf/MEK, JAK/STAT, PI3K/AKT, as well as epigenetic regulators such as histone acetyltransferases, histone deacetylases, etc. [55] As CHD4/NuRD complex functions as a chromatin remodeler that regulates the accessibility of variable complexes related to gene transcription, and a chromodomain motif in CHD4 does not directly bind to a specific DNA-sequence, CHD4 is more likely to regulate MDR1 expression via complex indirect transcriptional regulation rather than functions as direct transcription factor or enhancer [14,56].Suppression of ERBB signal cascade by CHD4 inhibition could be an example of several potent mechanisms [57].

Ovarian cancer is now recognized as a heterogeneous disease [28,58]. The genomic alteration profiles are quite distinct depending on histologic subtypes or even among the same subtypes [59]. As we observed increased platinum sensitivity by CHD4 knockdown consistently in multiple ovarian cancer cells derived from HGSC and clear cell carcinoma, it is plausible to consider that CHD4 suppression can be beneficial to the majority of ovarian cancer cases regardless of the subtype or genomic alterations. However, it is important to note that Guillemette et al. reported that CHD4 conferred cisplatin resistance in BRCA2-mutant cancer [60]. As BRCA2 mutation is found in 11–30% of HGSC, the indication of CHD4 inhibition in combination with cisplatin might be genetically tailored [61,62].

We have successfully demonstrated the potential of the novel SMARCA5/CHD4 inhibitor ED2-AD10 in combination with cisplatin for the first time in ovarian cancer cells. Along with the emerging literature reporting CHD4 as a candidate therapeutic target, our findings highlight the importance of developing clinically available CHD4 inhibitors.

A major limitation of this study is that we could not assess whether CHD4 suppression actually sensitizes clinically chemo-resistant cancer to platinum agents. Although it is difficult to evaluate this aspect appropriately with conventional cancer cell models, patient-oriented models such as patient-derived xenograft mice or organoids established from clinically proven platinum-resistant tumor tissue will help us address this question in the future. Another possibility is that the combination therapy with CHD4 inhibitor might be able to make platinum agents effective at lower concentrations, resulting in reduction of adverse events, if not overcoming resistance.

## Conclusions

In conclusion, we clarified the role of CHD4 in regulation of platinum sensitivity in ovarian cancer cells. We further showed the potential of CHD4 as a candidate therapeutic target in combination with platinum agents for ovarian cancer. Although the feasibility should be assessed further, CHD4 has the potential to bring benefits to patients with ovarian cancer.

## Supporting information

**S1 Fig. Correlation between CHD4 expression and clinical outcome.** a) Survival comparison between the two groups divided at median CHD4 mRNA expression. No statistically significant difference was observed. b) Representative examples positive and negative for CHD4 immunohistochemistry. A sample was considered to be CHD4 positive if more than 50% of the cancer cell nuclei were stained. A summary of CHD4 immunohistochemistry among the cases shown in Fig 1B. p = 0.28 by Fisher's exact test.
(DOCX)

**S2 Fig. CHD4 expression across ovarian cancer cell lines.** a) Quantitative PCR to compare CHD4 mRNA expression across cell lines. mRNA expression was normalized to GAPDH mRNA expression. b) Western blotting to compare CHD4 protein expression across cell lines.
(DOCX)

**S3 Fig. Cell viability assay with cisplatin treatment under CHD4 knockdown.** After 72 hours of siRNA transfection, TOV21G, JHOC5, and KURAMOCHI were treated with 0.5, 1, 2, 5, 10 μM of cisplatin. Relative cell viability was determined in 72 hours of cisplatin treatment and compared between CHD4 knockdown and control groups. $^*$p < 0.05; $^{**}$p < 0.01; $^{***}$p < 0.001. siCTRL; negative control siRNA.
(DOCX)

**S4 Fig. Clonogenic assay to assess the influence of CHD4 knockdown on cisplatin sensitivity.** Twenty-four hours after the transection of CHD4 siRNA or negative control siRNA, TOV21G cells were harvested and re-plated on a well of 6-well plate. The cells were treated with 5 μM of cisplatin or vehicle 24 hours after re-plating and colonies were stained 72 hours after cisplatin treatment. siCTRL, negative control siRNA.
(DOCX)

**S5 Fig. Dose-response curves to cisplatin treatment under CHD4 knockdown in OSE2 and OSE4.** Following 24 hours after transfection of control and CHD4 siRNA, OSE2 and OSE4 cells were treated by cisplatin for 72 hours, and then cell viability was assessed using Cell counting kit 8. IC50 of control and CHD4 knocked-down cells are 7.54 μM and 5.65 μM, respectively, in OSE2, and 2.37 μM and 2.23μM, respectively, in OSE4. IC50 values were compared between control and CHD4 knocked-down cells using Student's $t$-test. The knockdown efficacy of CHD4 was validated by western blotting. siCTRL, negative control siRNA.
(DOCX)

**S6 Fig. The influence of CHD4 knockdown on cell proliferation in ovarian cancer cells.** siRNA targeting CHD4 or negative control was transfected to two ovarian cancer cells derived from clear cell carcinoma subtype (TOV21G, JGOC5) and two high-grade serous subtype cells (KURAMOCHI, JGOS2). Cell viability was assessed 72, 96, and 120 hours after transfection. $^*$ represents p ≤ 0.05; Abbreviations: siCTRL, negative control siRNA; CCC, clear cell carcinoma; HGSC, high-grade serous carcinoma.
(DOCX)

**S7 Fig. Dose response results of ovarian cancer cell lines to ED2-AD101.** TOV21G, JHOC5, KURAMOCHI, and JHOS2 cells were treated with up to 30 μM of ED2-AD101. Each curve represents the mean of the triplicates.
(DOCX)

**S8 Fig. Influence of cisplatin treatment and CHD4 knockdown on cell cycle.** 72 hours after the transfection with CHD4 siRNA or negative control siRNA, TOV21G cells were treated

with 5 μM of cisplatin or vehicle for 24 hour and subjected to flow cytometry. The mean of three independent assays were documented. siCTRL; negative control siRNA.
(DOCX)

**S9 Fig. RAD51, p21, and p-glycoprotein expression under CHD4 knockdown.** (A) RAD51 and p21 expression 72 hours after siRNA transfection in JHOC5. Relative CHD4 expression normalized by the expression of β-actin was also determined by densitometry and demonstrated in the figure. (B) Quantification of RAD51 expression in TOV21G under CHD4 knockdown in comparison with control siRNA transfection. The bands were quantified by densitometry and CHD4 expression was normalized by β-actin expression. The data in the figure represents the mean of three independent assays. Statistical comparison was performed by one-sample t test. (C) MDR1 expression 72 hours after siRNA transfection in JHOC5 and JHOS2. Data were the representative of at least two independent assays. abbreviations: siCTRL; negative control siRNA. n.s.; not significant.
(DOCX)

**S10 Fig. Raw data of Western blotting for the figures and the supplementary information.**
(DOCX)

**S1 Table. IC50 determined by the dose-response results shown in Fig 2.**
(DOCX)

## Acknowledgments

The authors would like to thank Dr. Saunthararajah for kindly providing ED2-AD101.
    We would also like to thank Editage (www.editage.com) for English language editing.

## Author Contributions

**Conceptualization:** Yoshiko Oyama, Shogo Shigeta, Hideki Tokunaga, Keita Tsuji, Masumi Ishibashi, Yusuke Shibuya, Muneaki Shimada, Jun Yasuda, Nobuo Yaegashi.

**Data curation:** Yoshiko Oyama, Shogo Shigeta, Hideki Tokunaga.

**Formal analysis:** Yoshiko Oyama.

**Funding acquisition:** Shogo Shigeta, Hideki Tokunaga, Keita Tsuji, Nobuo Yaegashi.

**Investigation:** Yoshiko Oyama, Shogo Shigeta.

**Methodology:** Yoshiko Oyama, Shogo Shigeta, Hideki Tokunaga, Masumi Ishibashi, Yusuke Shibuya.

**Project administration:** Hideki Tokunaga, Nobuo Yaegashi.

**Resources:** Yoshiko Oyama, Shogo Shigeta, Hideki Tokunaga, Keita Tsuji, Masumi Ishibashi, Muneaki Shimada.

**Supervision:** Shogo Shigeta, Hideki Tokunaga, Yusuke Shibuya, Muneaki Shimada, Jun Yasuda, Nobuo Yaegashi.

**Validation:** Yoshiko Oyama, Hideki Tokunaga, Masumi Ishibashi.

**Visualization:** Yoshiko Oyama.

**Writing – original draft:** Yoshiko Oyama, Shogo Shigeta, Hideki Tokunaga.

**Writing – review & editing:** Shogo Shigeta, Hideki Tokunaga.

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
