## [Decision Letter · Decision Letter 0]

14 Dec 2020

PONE-D-20-33496

CHD4 regulates platinum sensitivity through MDR1 expression in ovarian cancer: A potential role of CHD4 inhibition as a combination therapy with platinum agents

PLOS ONE

Dear Dr. Tokunaga,

Thank you for submitting your manuscript to PLOS ONE. After careful consideration, we feel that it has merit but does not fully meet PLOS ONE’s publication criteria as it currently stands. Therefore, we invite you to submit a revised version of the manuscript that addresses the points raised during the review process.

This work is of relevance because it involved a new pathway likely relevant in the development of platinum resistance.

The reviewers concord in that it is not clear whether the differences in platinum sensitivity between clear cell vs. high-grade serous ovarian cancer is reflected in the set of cell lines selected. The toxicity of the novel compound ED2-AD101 (SMARCA5/CDH4 dual inhibitor) needs to be tested in non cancerous cells.Cisplatin sensitivity studies should be done over a concentration range of 1-10 micromolar. 

We look forward to receiving your revised manuscript.

Kind regards,

Carlos Telleria, PhD

Academic Editor

PLOS ONE

Journal Requirements:

2. In the ethics statement in the manuscript and in the online submission form, please provide additional information about the patient records/samples used in your retrospective study, including:

a) whether all data were fully anonymized before you accessed them;

b) the date range (month and year) during which patients' medical records/samples were accessed.

3. Please provide additional information about each of the cell lines used in this work, including any quality control testing procedures (authentication, characterisation, and mycoplasma testing). For more information, please see http://journals.plos.org/plosone/s/submission-guidelines#loc-cell-lines

4. Please provide accession numbers and/or URLs for the data obtained from the TCGA database.

5. Please provide the primer sequences used for RT-PCR.

6. Please note that PLOS does not permit references to “data not shown.” Authors should provide the relevant data within the manuscript, the Supporting Information files, or in a public repository. If the data are not a core part of the research study being presented, we ask that authors remove any references to these data.

7. PLOS ONE now requires that authors provide the original uncropped and unadjusted images underlying all blot or gel results reported in a submission’s figures or Supporting Information files. This policy and the journal’s other requirements for blot/gel reporting and figure preparation are described in detail at https://journals.plos.org/plosone/s/figures#loc-blot-and-gel-reporting-requirements and https://journals.plos.org/plosone/s/figures#loc-preparing-figures-from-image-files. When you submit your revised manuscript, please ensure that your figures adhere fully to these guidelines and provide the original underlying images for all blot or gel data reported in your submission. See the following link for instructions on providing the original image data: https://journals.plos.org/plosone/s/figures#loc-original-images-for-blots-and-gels.

8. Please include captions for your Supporting Information files at the end of your manuscript, and update any in-text citations to match accordingly. Please see our Supporting Information guidelines for more information: http://journals.plos.org/plosone/s/supporting-information

Additional Editor Comments:

The results presented in this article were found to be of interest to the field of ovarian cancer and demonstrate potential from a therapeutic standpoint. However, the reviewers felt that essential revisions need to be made to make this work publishable in the Journal.

Reviewers' comments:

Reviewer's Responses to Questions

**Comments to the Author**

1. Is the manuscript technically sound, and do the data support the conclusions?

Reviewer #1: Partly

Reviewer #2: Partly

Reviewer #3: Yes

2. Has the statistical analysis been performed appropriately and rigorously? 

Reviewer #1: Yes

Reviewer #2: I Don't Know

Reviewer #3: Yes

3. Have the authors made all data underlying the findings in their manuscript fully available?

Reviewer #1: Yes

Reviewer #2: Yes

Reviewer #3: Yes

4. Is the manuscript presented in an intelligible fashion and written in standard English?

Reviewer #1: Yes

Reviewer #2: No

Reviewer #3: Yes

5. Review Comments to the Author

Reviewer #1: In the manuscript, “CHD4 regulates platinum sensitivity through MDR1 expression in ovarian cancer: A potential role of CHD4 inhibition as a combination therapy with platinum agents”, the authors have characterized the role of CHD4 in sensitizing ovarian cancer cells towards platinum. The authors found that CHD4 expression is increased in platinum resistant clinical samples and knockdown of CHD4 enhanced cisplatin induced apoptosis. Furthermore, they found that CHD4 regulates platinum sensitivity via MDR1 expression. The findings are interesting and demonstrate great therapeutic potential in terms of combination therapy, however, the authors should address following concerns:

1) Have the authors compared CHD4 expression in platinum sensitive and platinum resistant ovarian cancer cell lines?

2) Supplementary Figure 3A: Knockdown of CHD4 is not convincing based on the immunoblotting for CHD4 and loading control. The authors should include a better representative blot along with densitometry analysis.

3) Figure 4F: Can the authors explain why there are two CDH4 bands in the empty vector lane?

4) Have the authors determined if cisplatin+CDH4 knockdown affect cell cycle progression?

5) Does treatment with cisplatin enhance the expression of MDR1 in ovarian cancer cells? Furthermore, does combination of CHD4 knockdown and cisplatin treatment affect MDR1 expression?

6) Can the authors speculate how CDH4 regulates MDR1 expression? What are the upstream regulators of MDR1? These points should be addressed in the discussion.

7) This study will greatly benefit by a phenotype rescue experiment. The authors can overexpress MDR1 in CHD4 knockdown cells and investigate if this changes cell viability or cisplatin induced apoptosis.

Reviewer #2: The goal of this study is to determine how overexpression of CHD4 contributes to platinum resistance in ovarian cancer.

The authors have genetically knocked down CHD4 expression in TOV21G cells and overexpressed CHD4 in the same cell line and determined cisplatin sensitivity using cell viability assays. It is not clear on what basis this cell line was chosen. Lack of a western blot showing the levels of CHD4 in the cell lines used is another major weakness. Were there any OV cell lines without CHD4 expression that the authors could have chosen to use for their overexpression studies.

Q-PCR analysis of what the authors define as cisplatin sensitive vs resistant clinical samples, the authors show that the resistant cells express higher levels of CHD4. Since the # of patient derived resistant samples is so few and the error bars overlap with the sensitive samples- the significance is not very clear. It is also not clear why the authors have not performed western blots for markers in the sensitive and resistant samples or by IHC for CHD4 and/or MDR1to claim CHD4 expression positively impacts the resistance to platinum treatment.

They also show synergy between SMARCA5/CHD4 dual inhibitor and cisplatin. Since ED2-AD101 is a novel drug, the author should show the cytotoxicity in the normal cells vs OC cells with positive control like 5FU with SRB or MTT and clonogenic assay and the same has to be performed in CHD4 knockdown cells with cisplatin also. Cisplatin sensitivity should be performed over a concentration range 0, 0.5, 1, 2,5,10 uM for better validation.

In fig 4, the authors show increased levels of both cleaved and non-cleaved PARP at 5uM of cisplatin and which makes it difficult to interpret these results. However, the authors revalidated the same in FACS analysis using a very high dose of cisplatin and claimed in CHD4 kd cells induces the cell death but author has not explained why there was an sudden increase in 5 to 20uM cisplatin concentration.

Although, the authors show in fig 4a, CHD4 kd with cisplatin induces phospho yH2AX, a marker for DNA damage and in ,Fig 4c CHD4 kd induces RAD51expression, a DNA repair factor, authors has to explain how cisplatin/ED2-AD101 inhibitor synergy will work on apoptosis?

Finally they have determined that MDR1 downregulation in CHD4 KD cells and overexpressed in cells with enhanced expression of CHD4.

Over all the study does not robustly add anything to the literature on the role of CHD4 due to the lack of well accepted methods to determine cell viability such as colony forming assays. It was very difficult to determine the IC50 values since the #s are not mentioned in the text. Lack of p value in Q-PCR based siRNA results made it difficult to assess the significance. In general the clarity of the figures also was not good. The manuscript is not acceptable for publication in PLOSONE with the limited amount of data presented in its current form.

Reviewer #3: Platinum drugs are the most effective in the ovarian cancer treatment. Platinum resistance is a critical issue to be solved. It is important to study the mechanism about platinum sensitivity or platinum resistance.　

The authors first found that CHD4 mRNA expression was significantly higher in the platinum resistant samples than in the platinum sensitive samples in ovarian cancer tissues. Then, the authors investigated whether CHD4 is associated with platinum sensitivity using ovarian cancer cell lines in vitro. Suppression of CHD4 expression increased the platinum sensitivity by decreasing MDR1 expression in ovarian cancer cells.

The present study clearly showed that CDH4 is associated with platinum sensitivity in ovarian cancer cells.

Major comments

1. Generally, the platinum sensitivity is much higher in high grade serous carcinoma than in clear cell carcinoma. According to Figure 2, there seems to be no difference in the platinum sensitivity between clear cell carcinoma cells (TOV21G and JHOC5) and high grade serous carcinoma cells (KURAMOCHI and JHOS2). Furthermore, the effect of CHD4 knockdown seems to be high in clear cell carcinoma cells compared with that in high grade serous carcinoma cells. Reviewer is wondering whether these cell-lines used in this study are suitable for evaluating platinum sensitivity in ovarian cancer cells.

2. Figure 3 only showed the data from clear cell carcinoma cell lines (TOV21G and JHOC5). The data from high grade serous carcinoma cell lines (KURAMOCHI and JHOS2) should be provided or described in the result section.

6. PLOS authors have the option to publish the peer review history of their article (what does this mean?). If published, this will include your full peer review and any attached files.

Reviewer #1: No

Reviewer #2: No

Reviewer #3: No

---

## [Author Response · Author response to Decision Letter 0]

25 Mar 2021

We are extremely thankful to the editor and reviewers for reviewing our manuscript and providing insightful comments. We have revised the manuscript based on all these comments, which have helped us improve our manuscript greatly. The revisions have been indicated as tracked changes in the revised manuscript.　We would like to inform the editor and reviewers that English proofreading was performed before resubmitting the manuscript, resulting in minor grammatical corrections as well as the revision in response to the reviewer’s comments. Below, we have provided point-by-point responses to each of the reviewers’ comments. Should there be any further concerns regarding our manuscript, we will gladly clarify. 

Reviewer #1: In the manuscript, “CHD4 regulates platinum sensitivity through MDR1 expression in ovarian cancer: A potential role of CHD4 inhibition as a combination therapy with platinum agents”, the authors have characterized the role of CHD4 in sensitizing ovarian cancer cells towards platinum. The authors found that CHD4 expression is increased in platinum resistant clinical samples and knockdown of CHD4 enhanced cisplatin induced apoptosis. Furthermore, they found that CHD4 regulates platinum sensitivity via MDR1 expression. The findings are interesting and demonstrate great therapeutic potential in terms of combination therapy, however, the authors should address following concerns:

1) Have the authors compared CHD4 expression in platinum sensitive and platinum resistant ovarian cancer cell lines?

　

Thank you for the valuable comment. To address your concern, we additionally compared CHD4 expression by western blotting in variable ovarian cancer cell lines including A2780 and A2780cis, a pair of parental cells and their derivatives that experimentally acquired platinum resistance. The result was demonstrated in S2 Fig, and additionally described in page 16, line 223-229. CHD4 expression did not differ across the cell lines. As there are plenty of mechanisms involved in platinum sensitivity, we considered that this result does not directly contradict our findings. We appreciate your suggestion.

2) Supplementary Figure 3A: Knockdown of CHD4 is not convincing based on the immunoblotting for CHD4 and loading control. The authors should include a better representative blot along with densitometry analysis.

Thank you for the valuable suggestion. We have replaced the bands with a better representative blot with the results of densitometry (S9 Fig A).

3) Figure 4F: Can the authors explain why there are two CDH4 bands in the empty vector lane?

Thank you for the valuable comment. The CHD4 antibody used in this study frequently detects two bands close to each other. In CHD4 knockdown, the expression of the lower band always decreased as shown in S5 Fig and S9 Fig. Hence, it is fair to conclude that the lower band represents CHD4. As the expression of the lower band obviously increases by the transfection of the CHD4 overexpression vector, we considered that overexpression of CHD4 was successfully introduced. 

4) Have the authors determined if cisplatin+CDH4 knockdown affect cell cycle progression?

Thank you for the valuable suggestion on cell cycle assessment. We performed this analysis additionally, and the result has been displayed in S8 Fig. The result has also been described in the results section of the revised manuscript. (page 21, line 312-318)

Briefly, cisplatin treatment alone facilitated S phase arrest in TOV21G. CHD4 suppression also increased the population of the cells at S phase. Mild increase of G2/M was also observed. A combination of CHD4 knockdown and cisplatin administration does not seem to interact with each other in the perspective of cell cycle regulation. 

5) Does treatment with cisplatin enhance the expression of MDR1 in ovarian cancer cells? Furthermore, does combination of CHD4 knockdown and cisplatin treatment affect MDR1 expression?

Thank you for the valuable comment. We examined the MDR1/p-glycoprotein expression by western blotting. It is shown in a figure 1 of the additional information for the reviewers. The expression of p-glycoprotein was not significantly affected by cisplatin treatment, CHD4 knockdown, or the combination. However, we could only get very faint bands for the p-glycoprotein once, after the replacement with a new one of a different lot.

6) Can the authors speculate how CDH4 regulates MDR1 expression? What are the upstream regulators of MDR1? These points should be addressed in the discussion.

Thank you for the valuable comment. Accordingly, we have addressed the points mentioned in this comment in the discussion section of the revised manuscript, with some relevant references. (page 25-26, line 392-401) 

7) This study will greatly benefit by a phenotype rescue experiment. The authors can overexpress MDR1 in CHD4 knockdown cells and investigate if this changes cell viability or cisplatin induced apoptosis.

We appreciate your suggestion. We transfected MDR1 overexpression vector by lipofection in TOV21G. Quantitative PCR analysis revealed the increase of MDR1 mRNA quickly disappeared within 72 hours and p-glycoprotein expression does not increase. These results have been demonstrated in a figure 2 of the additional information for the reviewers. We concluded that it is not possible to address the reviewer’s comment with transient vector transfection. As our laboratory currently does not have permission to handle viral vectors, we could not proceed to create a stable overexpression system. 

Reviewer #2: The goal of this study is to determine how overexpression of CHD4 contributes to platinum resistance in ovarian cancer.

The authors have genetically knocked down CHD4 expression in TOV21G cells and overexpressed CHD4 in the same cell line and determined cisplatin sensitivity using cell viability assays. It is not clear on what basis this cell line was chosen. Lack of a western blot showing the levels of CHD4 in the cell lines used is another major weakness. Were there any OV cell lines without CHD4 expression that the authors could have chosen to use for their overexpression studies.

Thank you for pointing out an important issue. We agree that it is important to select the cell lines based on the CHD4 expression level. We have added a supplementary figure (S2 Fig) to demonstrate the expression of CHD4 in several ovarian cancer cells lines including the cells used in this study. There was no clear difference in CHD4 expression across the cell lines.

From the perspective of translational research, we believe it is also important to select the cells that retain the characteristics unique to each histologic subtype as ovarian cancer is quite heterogenous. As there was no clear difference in CHD4 expression in ovarian cancer cells, we decided to mainly use TOV21G, JHOC5, and KURAMOCHI, JHOS2 that are reported to appropriately reflect the phenotype of clear cell carcinoma and high-grade serous carcinoma, respectively.　We have added several comments on this point in the revised manuscript. (page 16, line 223-231)

References

Domcke S, Sinha R, Levine DA, Sander C, Schultz N. Evaluating cell lines as tumour models by comparison of genomic profiles. Nat Commun. 2013;4: 2126. 

Beaufort CM, Helmijr JC, Piskorz AM, Hoogstraat M, Ruigrok-Ritstier K, Besselink N, et al. Ovarian cancer cell line panel (OCCP): clinical importance of in vitro morphological subtypes. PLoS One. 2014;9: e103988. 

Anglesio MS, Wiegand KC, Melnyk N, Chow C, Salamanca C, Prentice LM, et al. Type-specific cell line models for type-specific ovarian cancer research. PLoS One. 2013;8: e72162. 

Q-PCR analysis of what the authors define as cisplatin sensitive vs resistant clinical samples, the authors show that the resistant cells express higher levels of CHD4. Since the # of patient derived resistant samples is so few and the error bars overlap with the sensitive samples- the significance is not very clear. It is also not clear why the authors have not performed western blots for markers in the sensitive and resistant samples or by IHC for CHD4 and/or MDR1to claim CHD4 expression positively impacts the resistance to platinum treatment.

Thank you for the valuable comment. As mentioned in the manuscript, we considered a specimen to be a platinum resistant sample only if the remaining tumor in the patient clearly showed progressive or non-regressive behavior toward subsequent chemotherapy. As the majority of the samples were collected in the primary debulking surgery and up to 80 % of ovarian cancer is initially sensitive to platinum-based chemotherapy, there were only 6 samples left in the resistant group. The error bars in the figure represent standard deviation; thus, overlapping of two error bars does not deny the presence of a statistically significant difference. 

IHC was additionally performed. The results are demonstrated in a supplementary figure (S1 Fig B and C) and described in the revised manuscript (page 15, line 214-219).

Although statistically not significantly different, the resistant samples were all positive for CHD4 immunostaining and all of the CHD4 negative samples were included in the platinum sensitive group. We believe that the results support our findings. 

As to western blotting, the amount of stored frozen samples was substantially not enough to extract protein and subject to western blotting. We appreciate your comments on our clinical data set.

They also show synergy between SMARCA5/CHD4 dual inhibitor and cisplatin. Since ED2-AD101 is a novel drug, the author should show the cytotoxicity in the normal cells vs OC cells with positive control like 5FU with SRB or MTT and clonogenic assay and the same has to be performed in CHD4 knockdown cells with cisplatin also. Cisplatin sensitivity should be performed over a concentration range 0, 0.5, 1, 2,5,10 uM for better validation.

Thank you for the valuable comment. Based on your comment, we additionally performed clonogenic assays with ED2-AD101 in multiple cell lines including non-cancerous ovarian surface epithelial cells OSE2 and OSE4, and the results are summarized in a figure 3 of the additional information for the reviewers. The response to ED2-AD101 varies depending on each cell line and suppression of colony formation by ED2-AD101 was not specific to ovarian cancer cells. As CHD4 knockdown alone did not drastically suppress cancer cell proliferation as shown in the supplementary figure (S6 Fig), the results obtained were unsurprising.

In contrast to ovarian cancer cell lines, cisplatin sensitivity was not significantly increased by CHD4 knockdown in either OSE2 or OSE4 (S5 Fig). The result indicates that an increase in the platinum sensitivity by CHD4 inhibition could be cancer-cell specific.

We additionally performed the cisplatin sensitivity assay at the indicated concentrations. CHD4 knockdown significantly increased platinum sensitivity in TOV21G, JHOC5, and KURAMOCHI. The result is illustrated in S3 Fig and described in the revised manuscript (page 16-17, line 238-241).

In fig 4, the authors show increased levels of both cleaved and non-cleaved PARP at 5uM of cisplatin and which makes it difficult to interpret these results. However, the authors revalidated the same in FACS analysis using a very high dose of cisplatin and claimed in CHD4 kd cells induces the cell death but author has not explained why there was an sudden increase in 5 to 20uM cisplatin concentration.

Thank you for the insightful comment. As we found that PARP cleavage was not an appropriate marker for the assessment of apoptosis in this experimental setting, we preliminarily performed an apoptosis assay by flow cytometry with 5 �M of cisplatin and observed little increase in Annexin-V positive fraction regardless of CHD4 knockdown. The preliminary data is shown in a figure 4 of the additional information for the reviewers. Thus, we determined to increase the cisplatin concentration to 20 �M and successfully detected the increase in annexin V positive cell fraction by cisplatin. As DNA double-strand breaks do not always result in apoptosis, we considered that it is plausible to see the increase of Annexin V positive cells only at higher concentrations of cisplatin. 

Taken together, we conclude that CHD4 knockdown enhances cisplatin-mediated DNA-double strand breaks and facilitates apoptotic changes under a certain degree of DNA double strand accumulation.

Although we have not shown the flow cytometry result with 5�M of cisplatin in our manuscript, as this is preliminary data, we are happy to add it as a supplementary figure if the reviewer recommends that it should be included.

Although, the authors show in fig 4a, CHD4 kd with cisplatin induces phospho yH2AX, a marker for DNA damage and in ,Fig 4c CHD4 kd induces RAD51expression, a DNA repair factor, authors has to explain how cisplatin/ED2-AD101 inhibitor synergy will work on apoptosis?

Thank you for the valuable suggestion. The experiment was repeated three times independently; however, a consistent trend was not observed. We also quantified the density of RAD51 and did not find a significant difference in this as well. This result has been depicted in a supplementary figure (S9 Fig B) and described in the revised manuscript. (page 23, line 347-350) 

Finally they have determined that MDR1 downregulation in CHD4 KD cells and overexpressed in cells with enhanced expression of CHD4.

Over all the study does not robustly add anything to the literature on the role of CHD4 due to the lack of well accepted methods to determine cell viability such as colony forming assays. It was very difficult to determine the IC50 values since the #s are not mentioned in the text. Lack of p value in Q-PCR based siRNA results made it difficult to assess the significance. In general the clarity of the figures also was not good. The manuscript is not acceptable for publication in PLOSONE with the limited amount of data presented in its current form.

Thank you for the valuable comments. We apologize that some fundamentally important information were missing in the original manuscript. In the revised manuscript, a supplementary table (S table) has been added to show the actual IC50 for each dose-response curve. The p-value in the quantitative PCR has also been added in the figure legend (page 18, line 260).

Further, the results of clonogenic assays for TOV21G and JHOC5 have also been added in a supplementary figure (S4 Fig) in addition to cell viability assay.

Regarding the clarity of the figures, we confirmed whether the resolution of the original data submitted by us complied with the requirements instructed by PLOS ONE. We also found that the resolution of the PDF file was much lower than the original data. Therefore, kindly enquire with the editor regarding the provision of original files submitted, if necessary.

Reviewer #3: Platinum drugs are the most effective in the ovarian cancer treatment. Platinum resistance is a critical issue to be solved. It is important to study the mechanism about platinum sensitivity or platinum resistance.　

The authors first found that CHD4 mRNA expression was significantly higher in the platinum resistant samples than in the platinum sensitive samples in ovarian cancer tissues. Then, the authors investigated whether CHD4 is associated with platinum sensitivity using ovarian cancer cell lines in vitro. Suppression of CHD4 expression increased the platinum sensitivity by decreasing MDR1 expression in ovarian cancer cells.

The present study clearly showed that CDH4 is associated with platinum sensitivity in ovarian cancer cells.

Major comments

1. Generally, the platinum sensitivity is much higher in high grade serous carcinoma than in clear cell carcinoma. According to Figure 2, there seems to be no difference in the platinum sensitivity between clear cell carcinoma cells (TOV21G and JHOC5) and high grade serous carcinoma cells (KURAMOCHI and JHOS2). Furthermore, the effect of CHD4 knockdown seems to be high in clear cell carcinoma cells compared with that in high grade serous carcinoma cells. Reviewer is wondering whether these cell-lines used in this study are suitable for evaluating platinum sensitivity in ovarian cancer cells.

We appreciate your comment about cell lines. As the phenotype of ovarian cancer is quite heterogenous according to each histologic subtype, it is important to select the cell lines which retain their original biomolecular phenotype. Several studies have pointed out that many of the ovarian cancer cell lines commercially available lost their original features after a certain period of time. Some cell lines have obviously different characteristics compared to the cell lines they were derived from. As for platinum sensitivity, we consider short-term in vitro assays do not always reflect the actual response in the clinical setting. Thus, we prioritized using the cell lines that well represent the features of high-grade serous carcinoma or clear cell carcinoma. We added descriptions of the cell lines in the revised manuscript. (page 16, line 223-231) 

2. Figure 3 only showed the data from clear cell carcinoma cell lines (TOV21G and JHOC5). The data from high grade serous carcinoma cell lines (KURAMOCHI and JHOS2) should be provided or described in the result section.

Thank you for your recommendation. We added the results of KURAMOCHI and JHOS2 in Fig 3 with some comments in the result section of the revised manuscript. (page 18, line 268-281) 

6. PLOS authors have the option to publish the peer review history of their article (what does this mean?). If published, this will include your full peer review and any attached files.

---

## [Decision Letter · Decision Letter 1]

20 Apr 2021

CHD4 regulates platinum sensitivity through MDR1 expression in ovarian cancer: A potential role of CHD4 inhibition as a combination therapy with platinum agents

PONE-D-20-33496R1

Dear Dr. Tokunaga,

We’re pleased to inform you that your manuscript has been judged scientifically suitable for publication and will be formally accepted for publication once it meets all outstanding technical requirements.

Kind regards,

Carlos Telleria, PhD

Academic Editor

PLOS ONE

Additional Editor Comments (optional):

The authors have responded mostly in its majority to the comments of the reviewers and is now found acceptable for publication. 

Reviewers' comments:

Reviewer's Responses to Questions

**Comments to the Author**

1. If the authors have adequately addressed your comments raised in a previous round of review and you feel that this manuscript is now acceptable for publication, you may indicate that here to bypass the “Comments to the Author” section, enter your conflict of interest statement in the “Confidential to Editor” section, and submit your "Accept" recommendation.

Reviewer #1: (No Response)

Reviewer #3: All comments have been addressed

2. Is the manuscript technically sound, and do the data support the conclusions?

Reviewer #1: Partly

Reviewer #3: Yes

3. Has the statistical analysis been performed appropriately and rigorously? 

Reviewer #1: Yes

Reviewer #3: Yes

4. Have the authors made all data underlying the findings in their manuscript fully available?

Reviewer #1: Yes

Reviewer #3: Yes

5. Is the manuscript presented in an intelligible fashion and written in standard English?

Reviewer #1: Yes

Reviewer #3: Yes

6. Review Comments to the Author

Reviewer #1: 1. The authors have determined CHD4 expression in platinum sensitive and platinum resistant ovarian cancer cell lines and claimed that CHD4 expression remains consistent in all the cell lines tested (Sup Fig S2B). According to the authors, the lower band corresponds to CHD4 protein. However, the band is very faint, and the loading control is inconsistent to draw any conclusions. In fact, if the loading control is normalized, the cisplatin resistant A2780cis cells might demonstrate lower CHD4 expression compared to the parental cell line.

2. The authors have performed clonogenic assay to determine the effect of CHD4 knockdown on cisplatin sensitivity. The results from this assay should be quantified in terms of number of colonies. However, the authors have stained the colonies in 72 hr which is a short time period to form visible colonies for quantification. Is there a reason, the authors have used a shorter time point?

Reviewer #3: The manuscript has been appropriately revised according to the reviewer's comment, and is now acceptable for publication.

7. PLOS authors have the option to publish the peer review history of their article (what does this mean?). If published, this will include your full peer review and any attached files.

Reviewer #1: No

Reviewer #3: No

---

## [Editor Report · Acceptance letter]

1 Jun 2021

PONE-D-20-33496R1 

CHD4 regulates platinum sensitivity through MDR1 expression in ovarian cancer: A potential role of CHD4 inhibition as a combination therapy with platinum agents 

Dear Dr. Tokunaga:

I'm pleased to inform you that your manuscript has been deemed suitable for publication in PLOS ONE. Congratulations! Your manuscript is now with our production department. 

Kind regards, 

on behalf of

Professor Carlos Telleria 

Academic Editor

PLOS ONE